# How scars shape the neural landscape: Key molecular mediators of TGF-β1's anti-neuritogenic effects

Kye-Im Jeon[1,2], Krystel R. Huxlin[1,2]*

1 The Flaum Eye Institute, University of Rochester, Rochester, NY, United States of America, 2 The Center for Visual Science, University of Rochester, Rochester, NY, United States of America

* khuxlin@ur.rochester.edu

**Data Availability Statement:** Raw data is available at https://figshare.com/articles/figure/How_scars_shape_the_neural_landscape_key_molecular_mediators_of_TGF-_1_s_anti-neuritogenic_effect/

## Abstract

Following injury to the peripheral and central nervous systems, tissue levels of transforming growth factor (TGF)-β1 often increase, which is key for wound healing and scarring. However, active wound regions and scars appear to inhibit process outgrowth by regenerating neurons. We recently showed that corneal wound myofibroblasts block corneal nerve regeneration *in vivo*, and sensory neurite outgrowth *in vitro* in a manner that relies critically on TGF-β1. In turn, delayed, abnormal re-innervation contributes to long-term sensory dysfunctions of the ocular surface. Here, we exposed morphologically and biochemically-differentiated sensory neurons from the ND7/23 cell line to TGF-β1 to identify the intracellular signals regulating these anti-neuritogenic effects, contrasting them with those of Semaphorin (Sema)3A, a known inhibitor of neurite outgrowth. Neuronal morphology was quantified using phase-contrast imaging. Western blotting and specific inhibitors were then used to identify key molecular mediators. Differentiated ND7/23 cells expressed neuron-specific markers, including those involved in neurite extension and polarization. TGF-β1 increased phosphorylation of collapsin response mediator protein-2 (CRMP2), a molecule that is key for neurite extension. We now show that both glycogen synthase kinase (GSK)-3β and Smad3 modulate phosphorylation of CRMP2 after treatment with TGF-β1. GSK-3β appeared to exert a particularly strong effect, which could be explained by its ability to phosphorylate not only CRMP2, but also Smad3. In conclusion, TGF-β1's inhibition of neurite outgrowth in sensory neurons appears to be regulated through a highly-conserved signaling pathway, which involves the GSK-3β/CRMP-2 loop via both canonical and non-canonical mechanisms. It is hoped that by defining the signaling pathways that control neurite outgrowth in wound environments, it will become possible to identify optimal molecular targets to promote re-innervation following injury.

## Introduction

Nerve injuries pose many challenges to patients, ranging from mild discomfort to life-long impairment due to pain, loss of sensitivity, motor function and/or autonomic control [1].

12363686 (DOI: https://doi.org/10.6084/m9.
figshare.12363686.v1).

**Funding:** This research was supported by the
National Eye Institute of the National Institutes of
Health (R01 EY015836 to KRH and K-IJ, and Core
Grant P30 EY001319 to the Center for Visual
Science) and an unrestricted grant to the University
of Rochester's Department of Ophthalmology from
the Research to Prevent Blindness Foundation. The
funders had no role in study design, data collection
and analysis, decision to publish, or preparation of
the manuscript.

**Competing interests:** The authors have declared
that no competing interests exist.

While adult peripheral nerves can regenerate, reinnervation does not always recapitulate the
original anatomic distributions, nor do all patients with injured nerves regain appropriate
functions [2, 3]. Successful structural and functional regeneration depends on many factors,
including the intrinsic growth capacity of the injured cells, the presence of a local, permissive
environment and appropriate axonal guidance cues [4–6].

Wound environments are not generally permissive to nerve regeneration. One factor that
contributes to this phenomenon is transforming growth factor (TGF)-β1, which is normally
present at low levels in intact peripheral and central nervous systems, but increases massively
both proximal and distal to injury sites [7, 8]. TGF-β1 controls master switches for key events
in extracellular matrix formation and wound healing [9, 10], *via* both canonical, Small Moth-
ers Against Decapentaplegic (Smad) signaling and noncanonical pathways [11]. The increased
expression of TGF-β1 after injury plays positive roles by regulating the immune response,
modulating neuronal phenotype and regulating expression of growth factors important for
neuronal repair [12–14]. However, TGF-β1 also stimulates the formation of fibrotic scars at
the lesion site; this can be problematic for nerve regeneration, as scar-forming cells can inhibit
regrowth, plasticity and recovery of damaged neurons [15–18]. Scar formation is particularly
detrimental in the cornea, where it decreases transparency and the ability to see. The cornea is
also interesting in the context of TGF-β1-regulated scarring, because it is the most densely
innervated peripheral tissue in the human body [19]. Corneal nerves arise from the trigeminal
ganglion, and as schematically represented in **Fig 1**, are distributed through the anterior half of
the stroma and the epithelium. They are predominantly mechano-sensory and nociceptive,
and serve to protect the eye from outside elements [20–22]. Although corneal nerves are part
of the peripheral nervous system and able to regenerate, wounding of the cornea often results
in abnormal reinnervation, with surprisingly serious and long-lasting effects [23–30]. We
recently showed that corneal myofibroblasts, which differentiate largely from stromal kerato-
cytes in and around the wound site [31], inhibit the regrowth of corneal nerves into the wound
area [16, 18]–**Fig 1**. Key to the present experiments, this effect was reproduced *in vitro*, when
corneal myofibroblasts were co-cultured with neurons derived from the ND7/23 cell line, they
inhibited neurite outgrowth [16]. ND7/23 cells are created by fusing N18tg2 mouse neuroblas-
toma cells and neonatal rat dorsal root ganglion cells [32], and they are often used as an
immortalized proxy for peripheral [including corneal] sensory neurons because they are read-
ily differentiated by addition of nerve growth factor (NGF) [33–35]. Differentiated ND7/23
neurons extend neurites and express molecular markers that identify them as A-fiber
mechano-sensors and C-fibers [36]–two major classes of sensory nerves found in the cornea
and other peripheral tissues. Importantly, TGF-β1 was shown to be both necessary and suffi-
cient for the anti-neuritogenic effect of myofibroblasts on differentiated ND7/23 cells [16],
allowing us to now use this growth factor as a surrogate [in place of myofibroblasts] to define
molecular pathways that inhibit neurite outgrowth in pure neuronal cultures (i.e., cell-
autonomously).

Neurites are projections from the filopodia or lamellipodia of a neuron, which can subse-
quently become polarized into an axon or dendrite [37, 38]. During their elaboration, neurites
undergo stereotypical changes [39] that rely on the response of the intracellular cytoskeleton
to a variety of extracellular cues, including secreted factors [38]. Microtubules, cylindrical
polymers composed of α- and β-tubulin heterodimers, are the primary cytoskeletal elements
forming the core of cylindrical neurites [40]. Collapsin-response mediator protein-2 (CRMP2)
functions as a carrier of free tubulin heterodimers [41], which it delivers to the assembly-plus
ends of nucleating sites in growing microtubules, consequently promoting acetylation and
neurite extension. We previously reported that TGF-β1-induced reduction in neurite out-
growth by ND7/23 cells is associated with increased phosphorylation of CRMP2 [16]. One

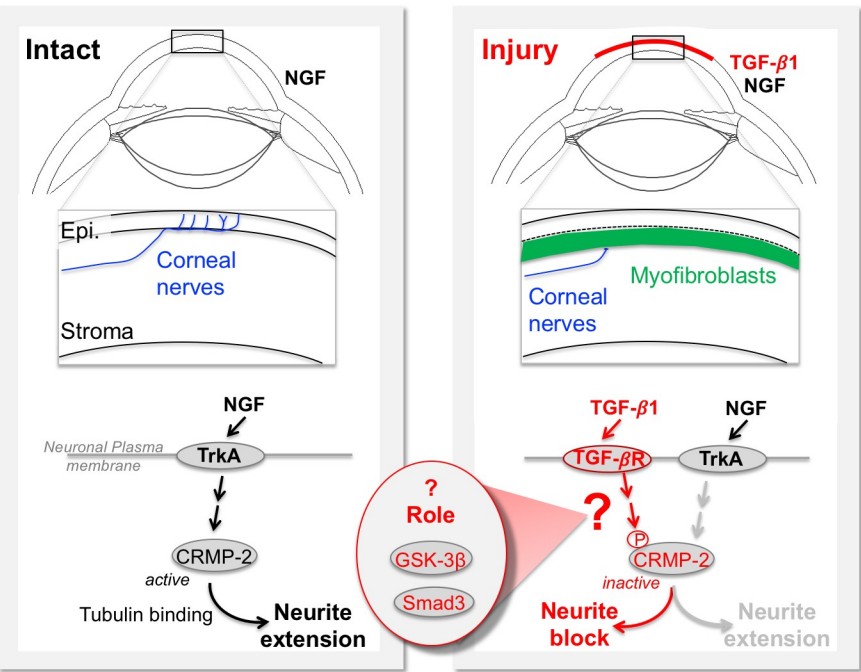

**Fig 1. Cellular and molecular events impacting nerves after injury of the anterior cornea of a large mammal.** The schematic illustrates specific gaps in our understanding of signaling pathways mediating the anti-neuritogenic effects of TGF-β1 on neurons addressed by the present experiments. Left panel: intact cornea. Right panel: damaged cornea.

class of molecules previously shown to phosphorylate CRMP2 in developing central neurons and a number of commercially-available neuronal cell lines is glycogen synthase kinase (GSK)-3β [42, 43]. GSK-3s are Ser/Thr protein kinases whose GSK-3α and GSK-3β isoforms are ubiquitously expressed [44] and play important roles in fundamental cellular processes such as differentiation, proliferation, motility and inflammation [45]. The present set of experiments asked if the inhibitory effects of TGF-β1 on neurite outgrowth occur through *activation* (i.e., de-phosphorylation) of GSK-3β and whether this is the key event causing increased CRMP2 phosphorylation, and the associated decreased neurite elongation (**Fig 1**). However, because TGF-β1 is well-known to exert many of its effects through canonical Smad signaling [11], we also investigated the role of this pathway in CRMP2 phosphorylation and GSK-3β activation (summarized schematically in **Fig 1**).

## Materials and methods

### Culture and priming of ND7/23 cells

The neuronal hybridoma cell line ND7/23 (Sigma Aldrich, St. Louis, MO) was maintained in Dulbecco's modified Eagle's medium (DMEM) (D6046; Sigma Aldrich) with 10% [vol/vol] of fetal bovine serum (FBS; Sigma Aldrich) + 1% [vol/vol] penicillin/streptomycin (Corning Inc., Corning, NY). Cells were multiplied until passage 10 (freshly thawed cells were considered passage 1). They were passaged by trypsinization (Corning Inc.) every 3–5 days, when approximately 70–90% confluent. Cells were seeded at a density of $5x10^4/cm^2$. After attachment, they were washed twice with 1x Dulbecco's phosphate-buffered saline (dPBS; Thermo Fisher Scientific, Waltham, MA) and with serum free medium (SFM) and then incubated with SFM for 1 day to inhibit mitosis before adding nerve growth factor (NGF) [46, 47]. Such SFM-primed cells were used in all experiments below, unless otherwise specified.

## Phenotypic characterization of SFM-primed ND7/23 cells after NGF treatment

Recombinant rat NGF (rNGF, 556-NG-100, R&D Systems Inc. Minneapolis, MN) was used to induce differentiation and neurite outgrowth in SFM-primed ND7/23 cells. SFM-primed ND7/23 cells ($6x10^5$ cells/6cm primaria culture dish, Corning Inc.) were treated with 50ng/ml rNGF + 0.5% FBS. Phase-contrast images of each culture dish were captured using an inverted Olympus IX73 microscope (Olympus Corporation of the Americas, Center Valley, PA) at 1, 3, 5 and 7 day (s). Cells were treated with fresh media containing 50ng/ml rNGF every second day. After imaging, western blotting was performed; whole cell lysates were fractionated by sodium dodecyl sulfate-polyacrylamide gel electrophoresis (SDS-PAGE) and transferred to a 0.2μm pore-size nitrocellulose membrane (Millipore-Sigma, Burlington, MA). After transfer, the membrane was cut into separate proteins according to molecular mass, and probed with antibodies against rabbit monoclonal growth associated protein-43 (GAP-43; 1:2000; #ab75810, Abcam, Cambridge, MA), mouse monoclonal neurofilament light polypeptide (NF-L; 1:1000; sc-20012, Santa Cruz Biotechnology, Dallas, TX), mouse monoclonal antibodies to anti-calcitonin gene-related peptide (CGRP; 1:500; C7113, Sigma Aldrich), rat monoclonal antibodies to substance P (SP, 1:500; MA5-17201, Thermo Fisher), and a mouse monoclonal antibody to acetylated α-tubulin (Ac-Tub; 1:5000, 6-11B-1, Santa Cruz Biotechnology). Responsiveness to NGF was assayed by probing for the expression of tropomyosin receptor kinase A (Trk-A; 1:2000; #06–574, Millipore-Sigma) [48]. Finally, we probed for changes in the expression of signaling molecules that could mediate the anti-neuritogenic effects of TGF-β1 on ND7/23 cells: total (t)-GSK-3α/β (mouse monoclonal antibody, 1:2000; sc-7291, Santa Cruz Biotechnology), and t-CRMP2 (mouse monoclonal antibody, 0.2ug-0.4ug/ml; C4G, Immuno-Biological Laboratories, Minneapolis, USA). The expression of β-actin (mouse monoclonal antibody, 1:10,000; sc-166729, Santa Cruz Biotechnology) was used as a loading control.

## Phenotypic characterization of ND7/23 cells after TGF-β1 treatment

To investigate the effects of TGF-β1 on neuritogenesis, we examined the morphology of ND7/23 cells after 1 day in culture, and measured their relative expression of pGSK-3β/tGSK-3α/β and pCRMP2/tCRMP2 using western blots. As a control, we contrasted the strength of TGF-β1' effects with those of Semaphorin 3A (Sema3A), a well-known inhibitor of neurite extension [49–51]. While earlier studies [49–51] found 4ng/ml Sema3A sufficient to produce 50% collapse of DRG growth cones, 10ng/ml of recombinant Sema3A was necessary to exert a comparable effect in SFM-primed ND7/23 cells and was thus used in all experiments.

**Quantitative analysis of neuronal polarity and neurite length.** SFM-primed ND7/23 cells ($8x10^4$ cells/6cm primaria culture dish) were pretreated with 1, 10 ng/ml TGF-β1 (240-B-010, R&D Systems) or 10ng/ml Sema3A (1250-S3-025, R&D Systems). After 1hr, either 0.5ng/ml or 50ng/ml rNGF (R&D Systems) was added for 1 day. Phase-contrast images were captured using an inverted Olympus IX73 microscope (Center Valley, PA) under 20X magnification. A 19-hole imaging template was used for analysis, covering ~8.6% of the dish area and ~100 cells/dish. The development of polarity by cultured cells can be divided into 5 stages [39, 40], but only stages 1 to 3 were evident after 1 day in our culture system (see examples in **Fig 3A–3C**). In Stage 1, short neurites emerge from the cell body, extending as protrusions and lamellipodia (lamellipodial veil). In Stage 2, some lamellipodia are replaced by short (less than 10μm) neurites, which appear as flattened processes with extensive, protrusive activity at their tips. In Stage 3, one of these neurites elongates at a faster rate, developing into an axon-like process (at least 10μm longer than the other neurites) [52], causing the neuron to become polarized. The CellSens imaging software (Olympus CellSens Standard ver. 1.12, Olympus)

was used to manually tag and count the number of cells in Stages 1, 2 and 3 in each of the 19 photographs taken for each culture plate. For each of the neurons in Stages 2 and 3, we then traced and measured the length of neurites extruded using the CellSens imaging software. From these tracing, we counted the number of neurites >40μm in length (two-fold longer than the average diameter of a cell soma) per image. These analyses allowed us to evaluate the neuronal differentiation attained following exposure to different factors, as well as the proclivity for significant (>40μm long) neurite extension.

**Effect of TGF-β1 on GSK-3β signaling in ND7/23 cells.** Passage 5, SFM-primed cells ($3x10^5$cells/35mm dish) were treated with 50ng/ml rNGF in 0.5% FBS-DMEM for 1 day. They were then washed with 1x dPBS and incubated in 0.5% FBS-DMEM. After 30min, cells were pretreated with either 10ng/ml of TGF-β1 or 10ng/ml Sema3A (as a positive control) for 30min before adding either 0.5ng/ml or 100ng/ml rNGF for 1hr. The cells were processed for western blots, which were stained using a rabbit polyclonal anti-pGSK-3β$_{Ser-9}$ antibody (1:2000; D85E12, Cell Signaling Technology) to detect and quantify the expression of p-GSK-3β relative to that of mouse monoclonal (total) t-GSK-3α/β (1:2000; SC-7291 Santa Cruz Biotechnology).

To determine if GSK-3β activity was a key regulator of CRMP2 phosphorylation, we used the natural GSK-3α/β inhibitor, lithium chloride (LiCl), to block its activity [53]. Passage 4–6, SFM-primed cells ($3x10^5$cells/35mm dish) were treated with 50ng/ml rNGF in 0.5% FBS-DMEM for 1 day. The cells were then washed and pretreated with 10ng/ml TGF-β1 or 10ng/ml Sema3A with/without 10mM LiCl (Sigma Aldrich) for 1hr. Finally, the cells were incubated with 100ng/ml rNGF for 1day before western blotting. A rabbit polyclonal anti-p-CRMP2$_{Thr-514}$ antibody (1:2000; STJ91107, St John's Laboratory Ltd, London, UK) was used to detect and quantify the expression of p-CRMP2 relative to that of total CRMP2 (see above, Immuno-Biological Laboratories).

**Role of Smad3 in TGF-β1/CRMP2 signaling in ND/23 cells.** While Smad2 and 3 are primary canonical mediators of TGF-β1's actions [11], Smad2 lacks a commercially-available, specific inhibitor. As such, we focused our studies on the contributions of Smad3 by employing SIS3 (Specific Inhibitor of Smad3), which specifically blocks activation of Smad3 [54]. Passage 4–6 SFM-primed cells ($3x10^5$cells/35mm dish) were incubated with 50ng/ml rNGF for 1day, then washed and pretreated with 10ng/ml TGF-β1 with/without 1μM of SIS3 (Sigma Aldrich) or 10 mM LiCl (Sigma Aldrich), the GSK-3β blocker, for 1hr. Higher concentrations of SIS3 (2.5, 5 and 10 μM) proved cytotoxic to ND7/23 cells cultured for 1 day and were therefore not used in the present experiments. Finally, 100ng/ml rNGF was added before harvesting and western blotting to check for the expression of p-GSK-3β/t-GSK-3α/β and of pCRMP2/β-actin (see above for antibodies used).

## Statistical analysis

When three or more intervention groups were compared, inter-group differences were assayed with one or two-way ANOVAs, followed by Tukey's post-hoc tests, as appropriate. When only two groups were compared, two-tailed paired or unpaired Student's t-tests were performed. A probability of error of $P<0.05$ was considered statistically significant.

## Results

### ND7/23 cells become neuron-like after sequential treatment with SFM and rNGF

When cultured with 10% FBS/DMEM, ND7/23 cells proliferated rapidly, with a strong tendency to form clumps that were loosely attached to the culture dish and occasionally became

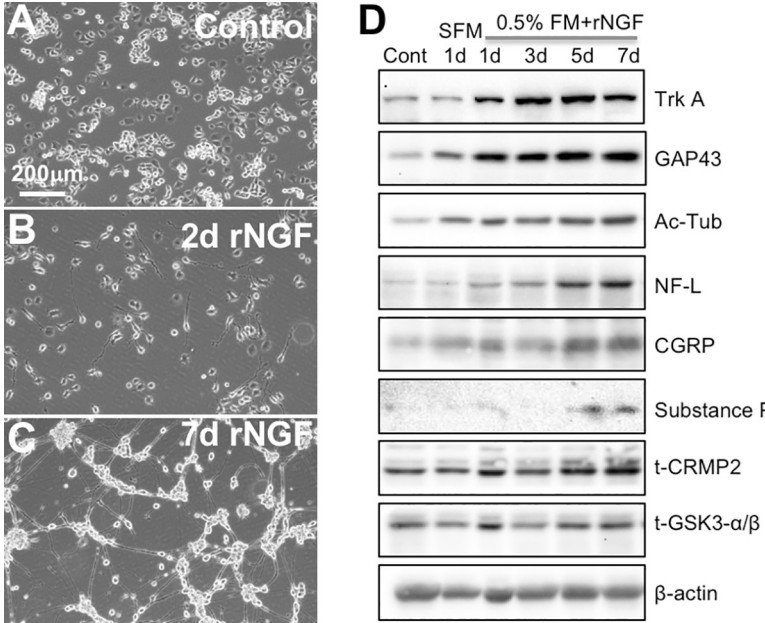

**Fig 2. Phenotypic characterization of differentiated ND7/23 cells. (A)** Phase-contrast image of ND7/23 cells cultured under 10% FBS/DMEM (control) conditions (see lane 1 in D). **(B)** Phase-contrast image of ND7/23 cells cultured in SFM (Serum-Free Medium) for 1 day, then re-plated with 0.5% FM (FBS+DMEM) + 50ng/ml rNGF for 1 day (lane 3 in D). **(C)** Phase-contrast photograph of ND7/23 cells cultured in SFM for 1 day, then re-plated in 0.5% FM + 50ng/ml rNGF for 7 days (lane 6 in D). Scale bar applies to all 3 photographs. **(D)** Representative Western blots of ND7/23 cells treated with 10% FBS/DMEM (lane 1, Cont.), or grown in SFM for 1 day (lane 2) or in 0.5% FM+50ng/ml rNGF for 1, 3, 5, or 7 days (lanes 3–6). The relative expression of TrkA, GAP43, acetylated α-tubulin (Ac-Tub), NF-L, CGRP, substance P, and t-CRMP2 all increased when cells were exposed to rNGF, albeit at different rates. t-GSK3 $\alpha/\beta$ expression appeared relatively stable. β-actin was used as a loading control.

free-floating (**Fig 2A**). When FBS was removed from the culture medium to block proliferation, the majority of the attached, clumped cells spread into a monolayer. After 1day of addition of rNGF, the cells began to display clear neurite outgrowth, with the majority belonging to the Stage 1 phenotype. Over ensuing days, an increasing proportion of ND7/23 cells transitioned to Stage 2 or Stage 3 morphologies (**Fig 2B**), similar to DRG neurons in culture [55]. After 7 days, ND7/23 cell cultures developed aggregations of rounded, phase-bright cell bodies and an extensive network of dendrites that spread over the culture dishes' surface between these aggregations (**Fig 2C**).

We then asked if the morphological changes observed in ND7/23 cells were accompanied by the appearance of neuron-specific markers and molecules involved in neurite extension and polarization. Compared to undifferentiated ND7/23 cells (grown in 10% FBS, lane 1, **Fig 2D**), culture in SFM for 1 day caused an upregulation of GAP-43, Ac-Tub and CGRP (lane 2, **Fig 2D**). Even more dramatic changes were seen in these molecules after rNGF was added (lanes 3–6, **Fig 2D**). The relative expression of the NGF receptor TrkA was only seen to increase after addition of rNGF (lanes 3–6, **Fig 2D**). NF-L and t-CRMP2 increased more slowly, peaking at 7 days in culture (lane 6, **Fig 2D**). Although substance P was not expressed by proliferating cells (lane 1, **Fig 2D**) and did not appear during the first 4 days of culture with rNGF (lanes 3–4, **Fig 2D**), it was induced in detectable amounts after 5 and 7 days in culture (lanes 5–6, **Fig 2D**). In contrast, the expression of t-GSK-3β and β-actin (the loading control) remained relatively unchanged throughout (lanes 1–6, **Fig 2D**).

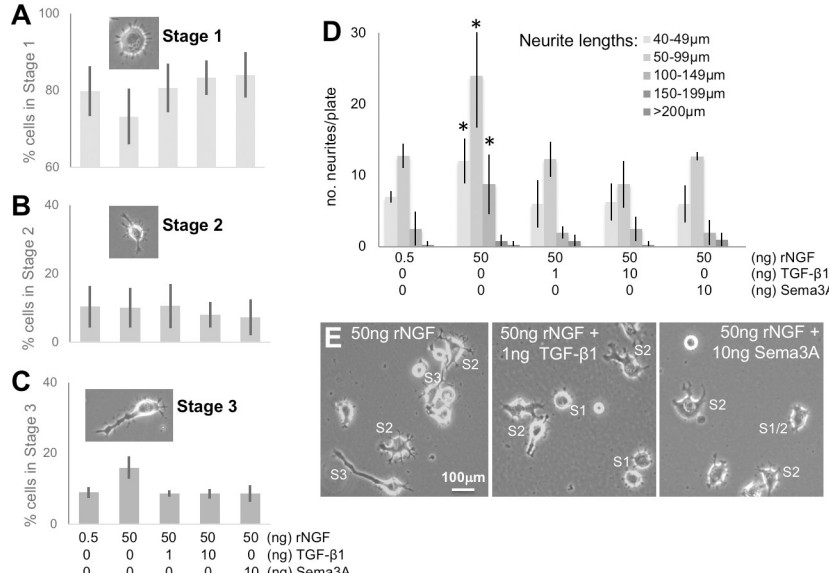

**Fig 3. Inhibitory effect of TGF-β1 on neurite extension in differentiated ND7/23 cells. (A)** Plot of percentage of cells counted that exhibited Stage 1 morphology when cultured with different concentrations of rNGF, TGF-β1 or Sema3A. Note the significant drop in the proportion of Stage 1 cells exposed to 50 ng/ml rNGF relative to all other conditions. Inset shown typical morphology of Stage 1 ND7/23 neuron. **(B)** Plot of percentage of cells counted that exhibited Stage 2 morphology under different culture conditions, as in A. The proportion of Stage 2 cells remained relatively unchanged across conditions. Inset shown typical morphology of Stage 2 ND7/23 neuron. **(C)** Plot of percentage of cells counted that exhibited Stage 3 morphology under different culture conditions, as in A and B. Inset shown typical morphology of Stage 3 ND7/23 neuron. Note the significant increase in the proportion of Stage 3 cells after exposure to 50 ng/ml rNGF and the lack of such increase in cells treated with NGF+TGF-β1 or Sema3A. **(D)** Plot of the number of neurites of different lengths per plate of ND7/23 cells grown under the same culture conditions as in A-C. Note the general increase in the number of neurites of all lengths >50μm long when cells are treated with 50 ng/ml rNGF, and the lack of such increase when the cells are treated with additional TGF-β1 or Sema3A. * p<0.05 relative to baseline (0.5ng/ml rNGF). **E.** Illustrations of phenomena shown in A-D using phase-contrast photographs of cultured ND7/23 cells treated either with 50 ng/ml rNGF, or with 50ng/ml rNGF+1 ng/ml TGF-β1, or with 50ng/ml rNGF+ 10ng/ml Sema3A. Scale bar applies to all 3 photographs. S1: Stage 1 cells, S2: Stage 2 cells, S3: Stage 3 cells. Graphs in A-D show means and standard deviations, with n = 4 in all cases except for the Sema 3A conditions, for which n = 3.

## Impact of TGF-β1 on neurite outgrowth

Having first established that differentiated ND7/23 cells express both TGF-βRl and TGF-βRll in their soma and neurites (**S1 Fig**), we then assessed how TGF-β1 impacted neurite outgrowth. When SFM-primed ND7/23 cells were cultured for 1 day with minimal (0.5 ng/ml) rNGF, the majority (~80%) of cells that extruded processes were in Stage 1 (**Fig 3A**), with the remaining 20% evenly divided between Stages 2 and 3 (**Fig 3B** and **3C**). If instead, these cells were cultured with 50ng/ml rNGF for 1 day (**Fig 3A–3C** and **3E**), the proportion of cells in Stage 1 decreased (two-tailed paired t-test for 0.5 *versus* 50 ng/ml rNGF conditions: $t_3 = 4.33$, $p = 0.023$), while the proportion of Stage 3 cells nearly doubled (two-tailed paired t-test: $t_3 = 3.58$, $p = 0.037$) The proportion of Stage 2 cells remained unchanged (two-tailed paired t-test: *ns*). Addition of 1, 10ng/ml TGF-β1 or 10ng/ml Sema3A to cultures exposed to 50ng/ml rNGF for 1 day maintained >80% of the cells in Stage 1 and kept the proportion in Stage 3 below 10%, in essence negating the impact of adding rNGF (**Fig 3A–3C** and **3E**). There were no significant differences between these 3 treatments (1, 10ng/ml TGF-β1 or 10ng/ml Sema3A) and the low rNGF condition: a two-way ANOVA with repeated measures on 1 factor (Stages) showed no significant effect of treatment ($F_3 = 0.28$, $p = 0.84$), a significant effect of Stage (1

*versus* 3, $F_1 = 1628.35$, $p<0.0001$) and no significant interaction between the two ($F_3 = 0.48$, $p = 0.703$). Finally, the number of neurites of lengths 40–149μm paralleled the proportion of cells in Stage 3 in each treatment group (**Fig 3D**). There were significantly more neurites 40–149μm in length when ND7/23 cells were cultured with 50 than with 0.5ng/ml rNGF; a two-way ANOVA with repeated measures on neurite length (40–49μm, 50–99μm or 100–149μm) showed a significant effect of neurite length category ($F_2 = 20.64$, $p = 0.00013$), of treatment ($F_1 = 27.42$, $p = 0.0019$) and no significant interaction ($F_2 = 1.32$, $p = 0.303$). Neurites longer than 150μm were very rare and their incidence appeared not to change dramatically under different treatment conditions (**Fig 3D**). Overall, the number of neurites >40μm went from 22.5 ±3.3/plate in 0.5ng/ml rNGF to 45.8±8.7/plate in 50ng/ml rNGF. Addition of 1, 10ng/ml TGF-β1 or 10ng/ml Sema3A to ND7/23 cells cultured with 50ng/ml rNGF negated the impact of rNGF on neurite outgrowth, with total number of neurites >40μm remaining around 20/plate, not significantly different than in the 0.5ng/ml rNGF condition (one-way ANOVA: $F_3 = 0.84$, $p = 0.4999$). All in all, TGF-β1 appeared to have a similar effect on neuritogenesis as Sema3A.

## Effect of TGF-β1 on intracellular signals associated with neurite outgrowth

Consistent with the notion that GSK-3β activity is a key regulator of NGF's neuritogenic effects [56] (**S2 Fig**), 100ng/ml rNGF dramatically increased levels of p-GSK-3β (the inactive form) while leaving levels of t-GSK-3 relatively unaffected (**Fig 4A and 4B**—lane2). As a result, there was a significant increase in the ratio of p-GSK-3β/t-GSK-3 after treatment with 50-100ng/ml rNGF (two-tailed Student's t-test: $t_2 = 4.48$, $p = 0.046$). Of note, the rNGF-induced increase in p-GSK-3β/t-GSK-3 ratio could be blocked effectively by the addition of 0.6μg/ml anti-NGF antibody (**S2 Fig**).

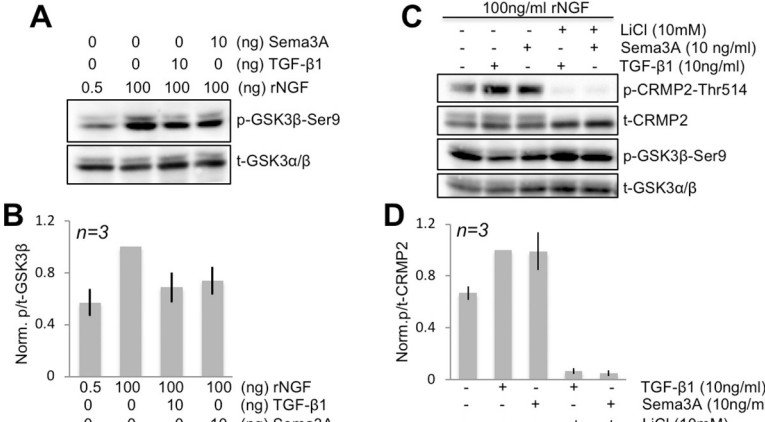

**Fig 4. Modulation of GSK3β activity by TGF-β1 and Sema3A. (A)** Representative Western blot of SFM-primed ND7/23 cells treated with different concentrations of rNGF with or without TGF- β1 or Sema3A and probed with antibodies against phosphorylated GSK-3β (at Ser9) or total GSK-3α/β. Levels of phosphorylated (i.e. inactive) GSK-3β (lane 2) are increased after treatment with 100ng/ml rNGF compared to 0.5 ng/ml rNGF (lane1). Both 10ng/ml TGF-β1 and 10ng/ml Sema3A blocked this increased phosphorylation of GSK-3β. **(B)** Plot of the ratio of p-GSK-3β/t-GSK-3α/β normalized to levels obtained following stimulation with 100ng/ml rNGF and under the different conditions shown in A. Data are means ± standard deviations. **(C)** Representative Western blot of SFM-primed ND7/23 cells treated with 100ng/ml rNGF alone (lane 1), or with TGF- β1 or Sema3A with and without lithium chloride (LiCl), a blocker of GSK-3β activity (see Fig 5). Both TGF- β1 and Sema3A cause an increase in levels of phosphorylated (p)-CRMP2 compared to total (t)-CRMP2 levels (compare lane1 with lanes 2,3). Pre-treatment with LiCl (lanes 4 and 5), which maximally phosphorylates (i.e. inactivates) GSK-3β prior to the addition of TGF- β1 or Sema3A completely blocks the ability of these two molecules to generate p-CRMP2. **(D)** Plot of the ratio of p-/t-cRMP2 measured from blots such as those shown in C and normalized to lane 2 (10ng/ml TGF- β1 condition). Values are means ± standard deviations.

Critical to our goals, addition of 10ng/ml TGF-β1 or 10ng/ml Sema3A (**Fig 4A and 4B**) to SFM -primed ND7/23 cells cultured with 100ng/ml rNGF were about equally effective at blocking the relative increase in p-GSK-3β. In fact, when the ratio of p-GSK-3β/t-GSK-3 was normalized to levels attained following stimulation with 100ng/ml rNGF, there was no significant difference between the effects of TGF-β1 and Sema3A (**Fig 4B**, two-tailed Student's t-test: $t_4 = 0.55$, $p = 0.612$), both causing about a 30% reduction. A one-way ANOVA comparing these 3 groups confirmed this, revealing a main effect of treatment ($F_{(2,4)} = 28.8$, $p = 0.00422$) with post-hoc Tukey HSD tests showing significant differences between rNGF and rNGF +TGF-β1 ($p<0.01$), as well as between rNGF and rNGF+Sema3A ($p<0.01$), but not between rNGF+TGF-β1 and rNGF+Sema3A.

Our prior works showed that TGF-β1 stimulation regulates phosphorylation of CRMP2 in ND7/23 cells [16]. Here, we replicated this result, showing that addition of 10ng/ml TGF-β1 or 10ng/ml Sema3A to SFM-primed ND7/23 cells cultured with 100ng/ml rNGF were equally effective at increasing levels of p-CRMP2, doing so by about 1.5-fold (**Fig 4C and 4D**). In fact, when the ratio of p-CRMP2/t-CRMP2 was normalized to levels attained following stimulation with 100ng/ml rNGF, there was no significant difference between the effects of 10ng/ml of TGF-β1 and 10ng/ml Sema3A (**Fig 4D**, two-tailed Student's t-test: $t_4 = 0.04$, $p = 0.970$).

Since GSK-3β is a candidate kinase thought to regulate CRMP2 phosphorylation and thus, its activity [42], the prediction is that when TGF-β1 and Sema3A decrease p-GSK-3β in ND7/23 cells, this should be associated with a corresponding increase in levels of p-CRMP2. Indeed, when SFM-primed ND7/23 cells were treated with 100ng/ml rNGF, the ratio of p-GSK-3β/t-GSK-3β increased (lane 2, **Fig 4A**, **and** **4B**), levels of p-CRMP2 were relatively low (lane 1, **Fig 4C and 4D**). In other words, CRMP2 was in its active (non-phosphorylated) form, associated with neurite extension (see Introduction, **Figs 1 and 6**). When either 10 ng/ml TGF-β1 or 10ng/ml Sema3A were added, the relative levels of p-GSK-3β decreased (**Fig 4A and 4B**) and those of p-CRMP2 increased (**Fig 3C and 3D**). Because levels of total GSK-3 and CRMP2 remained relatively unchanged, we concluded that an inverse relationship exists between levels of p-GSK-3β and p-CRMP2 in differentiated ND7/23 cells.

We then critically tested this hypothesis using lithium chloride (LiCl), a natural GSK-3α/β inhibitor, which is thought to work by activating an upstream protein kinase (AKT) and blocking protein phosphatases—actions that ultimately saturate the phosphorylation of GSK-3α/β on Ser 21/9 [53]. We verified that pre-treatment of ND7/23 cells with 10mM LiCl blocked the ability of both TGF-β1 and Sema3A to convert p-GSK-3β to GSK-3β; the end result was a maintenance of high levels of p-GSK-3β/t-GSK-3 in the cells (lanes 4 and 5, **Fig 4C**). Associated with this effect, LiCl completely blocked the ability of both TGF-β1 and Sema3A to increase p-CRMP2 relative to t-CRMP2 (compare lane 2–3 with lanes 4–5, **Fig 4C and 4D**). A one-way ANOVA for p-/t-CRMP2 ratios across all treatment groups in this experiment showed a significant main effect of treatment ($F_{(4,8)} = 8.3$, $p = 0.006$), with post-hoc Tukey HSD tests confirming significant differences between TGF-β1, Sema3A and TGF-β1+LiCl, Sema3A+LiCl conditions ($p<0.05$ in all cases). Therefore, it appears that levels of p-GSK-3β and p-CRMP2 are inversely and causally linked in ND7/23 cells, and that manipulations which either increase (e.g. LiCl) or decrease (e.g. TGF-β1, Sema3A) p-GSK-3β levels impact p-CRMP2.

## Relative importance of Smad3 for TGF-β1's effect on p-CRMP2

Since Smad2/3 signaling is a canonical mediator of TGF-β1's actions, we asked what role this pathway may play in TGF-β1's ability to modulate levels p-CRMP2 and p-GSK-3β in differentiated ND7/23 cells. First, we noted that TGF-β1 and NGF appeared to influence different

phosphorylation sites on Smad3 (compare lanes 1 and 2, **S3 Fig**). For instance, TGF-β1 appeared to modulate phosphorylation of Smad3 on Ser-204, but not on Ser-423/425. Furthermore, the Smad3 inhibitor SIS3 blocked Smad3 phosphorylation by TGF-β1 but not NGF (compare lanes 2 and 3, **S3 Fig**). LiCl, which blocks GSK-3β activity, blocked phosphorylation of Smad3 on both Ser-423/425 and Ser-204, while application of SIS3 failed to alter levels of p-GSK-3β (lanes 1–3, **Fig 5A and 5C**; see also **S4 Fig**); however, SIS3 did prevent TGF-β1 from raising p-CRMP levels above baseline (lanes 1–3, **Fig 5A and 5B**). In fact, a two-tailed t-test revealed no significant difference in the levels of p-CRMP2/β-actin in ND7/23 cells at baseline (i.e., cultured in 100ng/ml rNGF) and after treatment with TGF-β1+SIS3 ($t_4 = 1.04$, $p = 0.357$). Nonetheless, the GSK-3β inhibitor LiCl exerted a much stronger inhibitory effect on p-CRMP2 than SIS3, causing levels p-CRMP2 to drop about 4-fold below baseline levels and about 7-fold below levels induced by TGF-β1 stimulation (lane 4, **Fig 5A and 5B**). A one-way ANOVA across all treatment groups shown in **Fig 5** revealed a significant main effect of treatment ($F_{(3,6)} = 15.55$, $p = 0.0031$) with post-hoc Tukey's tests confirming significant differences in normalized p-CRMP2 levels between baseline (100ng/ml rNGF) and TGF-β1+LiCl ($p < 0.05$), as well as between TGF-β1 and TGF-β1+SIS3 ($p < 0.05$) and between TGF-β1 and TGF-β1+LiCL ($p < 0.01$). In summary, canonical Smad2/3 signaling does appear to contribute to TGF-β1's anti-neuritogenic effect in ND7/23 cells, but its contribution is smaller than that from GSK-3β.

## Discussion

In the present study, we developed an optimized protocol to generate morphologically and biochemically-differentiated sensory neurons from the ND7/23 cell line. We then used these differentiated neurons to detail the signaling pathways mediating the inhibitory effects of TGF-β1 on neurite outgrowth. Specifically, we uncovered that TGF-β1 works via both canonical and non-canonical signals, with the latter appearing to exert the strongest influence on CRMP2 phosphorylation, and consequently, neurite elongation.

### ND7/23 cells become neuron-like after sequential treatment with SFM and rNGF

Because our experiments were performed in ND7/23 cells, an immortalized cell line often used as an proxy for peripheral sensory neurons [16, 33–36], it was essential to first attain reliable differentiation of these cells. Neuroblastoma cells express few NGF receptors and can thus have low sensitivity to this growth factor [57, 58]. We overcame this hurdle by first "starving" ND7/23 cells with SFM for 1 day in order to increase responsiveness to NGF [55]. As a result, cells began to reliably express high levels of TrkA upon subsequent exposure to NGF, sustaining those levels across multiple (up to 7) days in culture. These cells also extended neurites, becoming multipolar (Stage 1), then pseudo-unipolar (Stages 2–3). Neurite extension was associated with increased acetylated α-tubulin (necessary for stable microtubule assembly), as well as NF-L. In addition, differentiated cells expressed neuronal markers such as GAP43 (associated with growing neurites), CGRP and substance P, which characterize major sub-populations of sensory nerves in the cornea [20]. Thus, in contrast with other neuroblastoma cells, which can extend neurites, but do not become polarized [59], we successfully developed culture conditions that allowed us to differentiate ND7/23 cells into mature, polarized, sensory, peripheral neurons.

Treatment of SFM-primed cells with 50ng/ml rNGF for 1 day decreased the proportion of Stage 1 cells and increased the proportion of Stage 3 cells by a corresponding amount; the proportion of Stage 2 cells remained unaffected. This suggests that NGF caused a proportion of

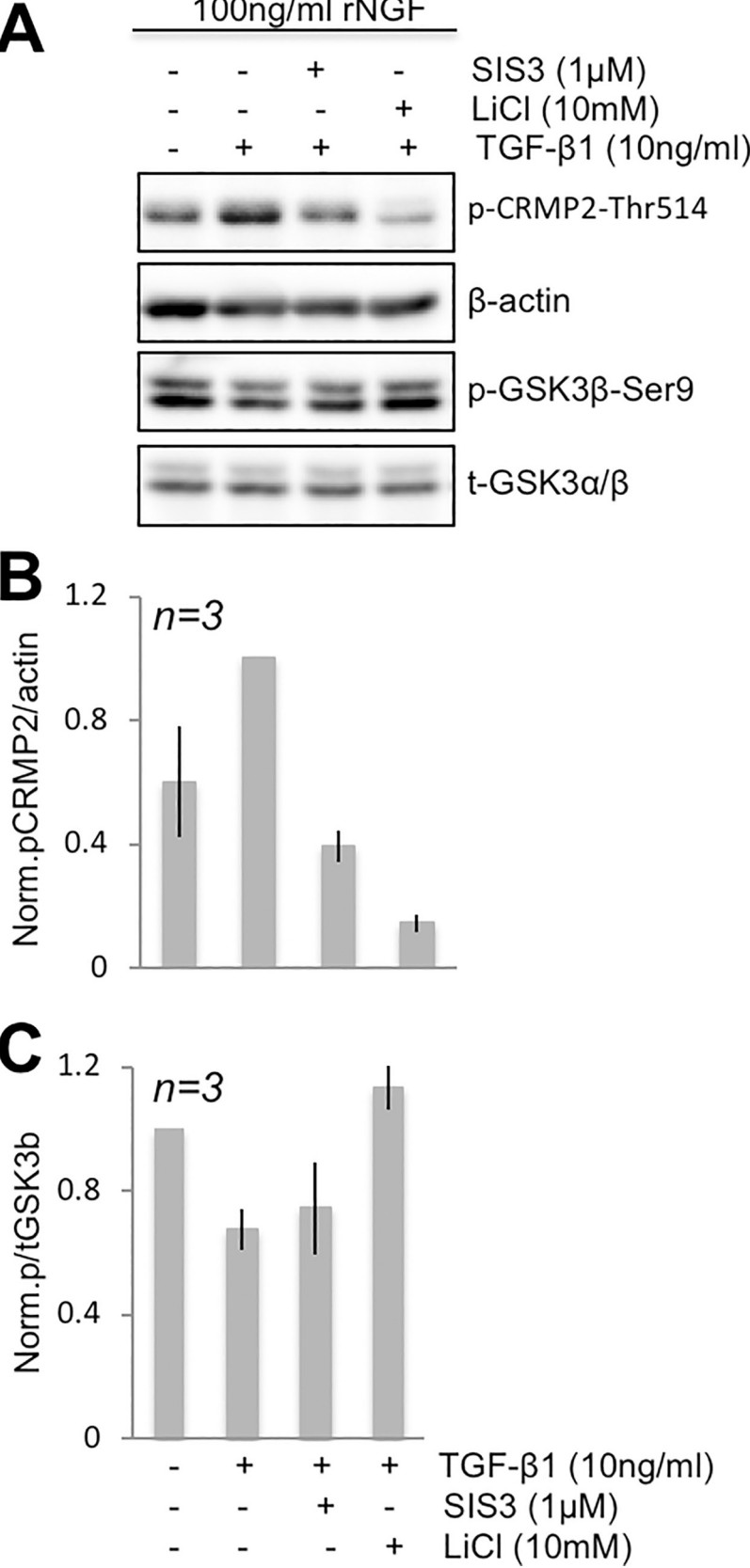

**Fig 5. SIS3 decreases phosphorylation of CRMP2 without activating GSK-3β. (A)** Representative Western blot of SFM-primed ND7/23 cells pretreated with SIS3 (lane 3) and LiCl (lane 4) with TGF- β1 and probed with antibodies against phosphorylated CRMP2 and GSK-3β (at Ser9) or total GSK-3α/β. Note that 1μM SIS3 significantly reduced levels of p-CRMP2 (lane 3) without affecting levels of p-GSK-3β. **(B)** Plot of the ratio of p-CRMP2/β-actin normalized to levels obtained following stimulation with 10ng/ml TGF-β1 (lane 2). β-actin was used as a loading control. **(C)** Plot of the ratio of p-GSK-3β/t-GSK-3α/β normalized to levels obtained following stimulation with 100ng/ml NGF (lane 1). Total GSK-3α/β was used as a loading control. All plotted data are means ± standard deviations over 3 experiments.

Stage 1 cells to reach Stage 2 and almost immediately transition into Stage 3, showing the greatest level of polarization with the extension of an axon-like process. Treatment of these cells with either TGF-β1 or Sema3A, even in the presence of 50ng/ml rNGF, prevented neurite elongation, and thus cell polarization. As such, there was no decrease in the proportion of Stage 1 cells, nor increase in Stage 3 cells. While we recently showed that ND7/23 cells co-cultured with corneal fibroblasts are sensitive to the effects of TGF-β1 [16], the cells in our prior publication were not SFM-primed, nor as carefully differentiated and staged as the cells in the present study. As such, we can now confidently state that TGF-β1 exert a strong, anti-neuritogenic effect on sensory neurons differentiated from the ND7/23 cell line.

## Molecular substrates of TGF-β1 's anti-neuritogenic effects in differentiated, sensory neurons

We began our investigations by ascertaining that SFM-primed, differentiated ND7/23 cells strongly express both TGF-βRl and TGF-βRll in their soma and neurites. This confirms a simple molecular initiator for TGF-β1 signaling in these neurons, as previously reported for primary sensory neurons [60], and in contrast with some neuroblastoma cells, whose low responsiveness to TGF-β1 was due to low levels of its receptor(s) [61]. Our earlier observation that the inhibitory effect of TGF-β1 on neurite outgrowth is critically mediated by its receptor [16] is consistent with studies in embryonic hippocampal and human iPSC-derived neurons

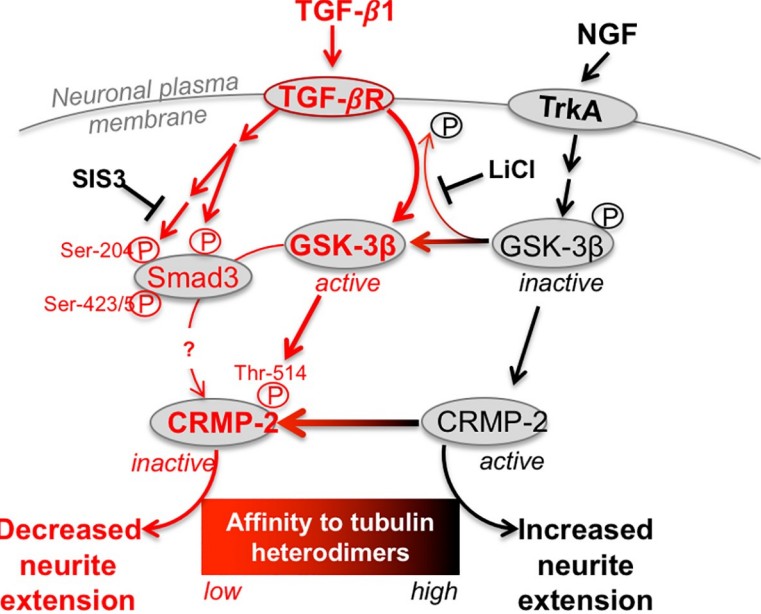

**Fig 6. Schematic diagram summarizing signaling pathways that mediate TGF-β1's inhibition of neurite outgrowth (in red).** The relative roles of GSK-3β and Smad3 in regulating CRMP2 activity are highlighted, as are the putative loci of action of LiCl on GSK-3β and of SIS3 on Smad3 in differentiated ND7/23 cells.

[62], in cerebellar granule neurons [63], in experimental diabetic neuropathy and in embryonic DRG cells [64]. However, TGF-β1 was reported to increase neurite length in differentiated retinal ganglion cells [65], suggesting that the effect of TGF-β1 on neuronal morphogenesis may be influenced by cell type, developmental state, health or disease, experimental manipulation, as well as by the local microenvironment. Nonetheless, most of the evidence in both central and peripheral nervous systems suggests that TGF-β1 has largely anti-neuritogenic effects, and this is also the case in differentiated neurons from the ND7/23 cell line. In fact, in these cells, present results show that TGF-β1 exerted the same level of inhibition on neurite outgrowth as Sema3A [49–51], at the same effective concentration (10ng/ml). So how does TGF-β1 exert its anti-neuritogenic effect in cultured, sensory neurons?

We previously showed that in ND7/23 cells, TGF-β1 increases phosphorylation of CRMP2 [16], a molecule that delivers free tubulin heterodimers to the assembly-plus ends of growing microtubules, thus promoting acetylation and neurite extension [41]. Here, we asked how TGF-β1 increased p-CRMP2, taking our cues from primary DRG cells, where Sema3A exerts is anti-neuritogenic effect by increasing GSK-3β activity, causing phosphorylation of CRMP2 [42, 66]. Conversely, neurotrophin-induced axonal outgrowth in hippocampal neurons occurs via decreased GSK-3β activity, which reduces phosphorylation of CRMP2 [42]. As such, it appears that in many neuronal types, CRMP2 is a physiological substrate of GSK-3β [67, 68] and that GSK-3β regulates neurite outgrowth through phosphorylation of CRMP2. In ND7/23 cells, treatment with either TGF-β1 or Sema3A prevented NGF-induced increases in p-GSK-3β levels to a similar extent. Because this caused a simultaneous increase in p-CRMP2, it suggests that modulating GSK-3β activity was key to regulating neurite outgrowth of differentiated ND7/23 cells (**Fig 6**). Our experiments are the first to show that TGF-β1 and Sema3A have comparable anti-neuritogenic effects—via GSK-3β/pCRMP2—in differentiated ND7/23 cells. Additionally, since the effects of TGF-β1 and Sema3A were similar in magnitude and operated via the same intracellular pathways in ND7/23 cells as in other neuronal cell types [42, 69, 70], we posit that neuritogenesis may be regulated through a very conserved signaling pathway that relies on a GSK-3β/CRMP-2 loop (**Fig 6**).

Finally, the actions of TGF-β1 in both neurons and other tissues of the body are known to be mediated by canonical Smad2/3 signaling [11]. Although not known to be regulated by TGF-β1, Smad1 signaling, which is key for regeneration of peripheral sensory neurons, is critically regulated by GSK-3β [71]. Thus, we also asked if TGF-β1 stimulation recruited Smads (specifically Smad3) to mediate its anti-neuritogenic effects in ND7/23 cells, and how this was related to its regulation of GSK-3β activity. SIS3 did indeed suppress TGF-β1-induced phosphorylation of Smad3 in these neurons, just as it does in fibroblasts [54]. However, here, this action prevented TGF-β1 from increasing levels of p-CRMP2, *without* affecting changes in p-GSK-3β. In contrast, LiCl, which blocks GSK-3β activity, decreased Smad3 phosphorylation at 2 different sites (impacted by TGF-β1 [72] and NGF, respectively). That GSK-3β can phosphorylate Smad3 at Ser-204 was previously reported [72] but its ability to also influence Smad3 phosphorylation at Ser-423/5 is newly described here. Thus, it appears that GSK-3β not only phosphorylates CRMP2 directly, it additionally phosphorylates (and activates) Smad3 (**Fig 6**). This explains why LiCl, TGF-β1 and other ways of modulating GSK-3β activity have such a strong effect on p-CRMP2 and neurite outgrowth. However, as Smad3 is not a kinase, how its phosphorylation causes increased levels of p-CRMP2 in differentiated ND7/23 cells, and whether it is sufficient and necessary, remain to be determined.

In conclusion, the present study used an optimized protocol to generate morphologically and biochemically-differentiated sensory neurons from the ND7/23 line. In these cells, we quantified the anti-neuritogenic effects of TGF-β1, contrasting them with those of Sema3A. We identified a key role of GSK-3β and p-CRMP2 in mediating the effects of both molecules.

The prevalence of a GSK-3β/CRMP2 loop for regulating neurite extension across a wide range of neuronal types and species, suggests this to be a highly-conserved signaling pathway. Ultimately, it also means that areas of central and peripheral nervous system damage, which are characterized by dramatically-increased levels of TGF-β1 relative to the healthy environment, provide a direct, highly-effective inhibitory stimulus to prevent neurons from rapidly innervating (or re-innervating) damaged regions. In the cornea and other tissues where scars form, the region of fibrosis is denuded of sensory nerve endings, and thus, of sensation. This may be necessary initially, to avoid the largely noxious stimulation that may result from exposure to molecules characteristic of the wound environment. However, one could also envisage a time when it would be desirable to restore normal innervation, a situation that also applies to the central nervous system where axons may need to regrow through glial scars. Thus, understanding key signaling pathways that control neurite outgrowth in wound environments, and the seemingly ubiquitous nature of these pathways across neuronal types, should allow us to better target interventions so as to promote re-innervation in a wide range of injury conditions. One option of course, is to pharmacologically block TGF-β1's binding to its receptor using for example, SB431542 [16]. While this works well to restore neurite outgrowth *in vitro*, the danger *in vivo* is that blocking all of TGF-β1's effects may inhibit beneficial functions of this growth factor in tissue repair. One alternative is to block CRMP2 phosphorylation by modulating the activity of GSK-3β in neurons. LiCl is well known as a neuro-active drug, and as a non-selective blocker of GSK-3β activity [53]. In the last decade, several selective, blood-brain-barrier penetrant GSK3 inhibitors have been described (reviewed in [73]Among them is SAR502250 (a.k.a. UDA-680), which was recently shown to exhibit neuroprotective activity in rodent model of Alzheimer's disease [74] Topical application of such pharmacologics to the cornea have not yet been attempted. Nor do we know if topical applications to the ocular surface can sufficiently target corneal nerve terminals, of whether more sophisticated, cell specific approaches (such as transfection with genetically-encoded materials–e.g. [75, 76] are needed to ultimately manipulate GSK-3β activity in damaged, corneal nerves.

## Supporting information

**S1 Fig. Differentiated ND7/23 cells express two different TGFβ receptor subunits.**
(DOCX)

**S2 Fig. NGF regulates activation (i.e. phosphorylation) state of GSK-3β in ND7/23 cells.**
(DOCX)

**S3 Fig. Effect of NGF, TGF-β1, SIS3 and LiCl on 2 different phosphorylation sites on Smad3.**
(DOCX)

**S4 Fig. Different concentrations of the Smad3 inhibitor SIS3 fail to impact levels of p-GSK-3β in ND7/23 cells.**
(DOCX)

**S1 Raw images.**
(PPTX)

## Acknowledgments

The authors wish to thank Thurma McDaniel for performing the immunohistochemistry in S1 Fig. We also thank Dr. Keith Nehrke, Thurma McDaniel, Margaret DeMagistris and Christine Callan for constructive comments on the manuscript.

## Author Contributions

**Conceptualization:** Kye-Im Jeon, Krystel R. Huxlin.

**Data curation:** Kye-Im Jeon, Krystel R. Huxlin.

**Formal analysis:** Kye-Im Jeon, Krystel R. Huxlin.

**Funding acquisition:** Krystel R. Huxlin.

**Investigation:** Krystel R. Huxlin.

**Methodology:** Kye-Im Jeon.

**Project administration:** Krystel R. Huxlin.

**Resources:** Krystel R. Huxlin.

**Supervision:** Krystel R. Huxlin.

**Validation:** Kye-Im Jeon.

**Visualization:** Kye-Im Jeon, Krystel R. Huxlin.

**Writing – original draft:** Kye-Im Jeon, Krystel R. Huxlin.

**Writing – review & editing:** Kye-Im Jeon, Krystel R. Huxlin.

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
