## [Decision Letter · Decision Letter 0]

1 Oct 2020

PONE-D-20-16873

How scars shape the neural landscape: key molecular mediators of TGF-β1’s anti-neuritogenic effects

PLOS ONE

Dear Dr. Huxlin,

Thank you for submitting your manuscript to PLOS ONE. After careful consideration, we feel that it has merit but does not fully meet PLOS ONE’s publication criteria as it currently stands. Therefore, we invite you to submit a revised version of the manuscript that addresses all comments and queries point-by-point raised during the review process.

We look forward to receiving your revised manuscript.

Kind regards,

Antal Nógrádi, M.D., Ph.D., D.Sc.

Academic Editor

PLOS ONE

Journal Requirements:

2.PLOS ONE now requires that authors provide the original uncropped and unadjusted images underlying all blot or gel results reported in a submission’s figures or Supporting Information files. This policy and the journal’s other requirements for blot/gel reporting and figure preparation are described in detail at https://journals.plos.org/plosone/s/figures#loc-blot-and-gel-reporting-requirements and https://journals.plos.org/plosone/s/figures#loc-preparing-figures-from-image-files. When you submit your revised manuscript, please ensure that your figures adhere fully to these guidelines and provide the original underlying images for all blot or gel data reported in your submission. See the following link for instructions on providing the original image data: https://journals.plos.org/plosone/s/figures#loc-original-images-for-blots-and-gels.

3.PLOS requires an ORCID iD for the corresponding author in Editorial Manager on papers submitted after December 6th, 2016. Please ensure that you have an ORCID iD and that it is validated in Editorial Manager. To do this, go to ‘Update my Information’ (in the upper left-hand corner of the main menu), and click on the Fetch/Validate link next to the ORCID field. This will take you to the ORCID site and allow you to create a new iD or authenticate a pre-existing iD in Editorial Manager. Please see the following video for instructions on linking an ORCID iD to your Editorial Manager account: https://www.youtube.com/watch?v=_xcclfuvtxQ

<h1>** **</h1>

Reviewers' comments:

Reviewer's Responses to Questions

**Comments to the Author**

1. Is the manuscript technically sound, and do the data support the conclusions?

Reviewer #1: Yes

Reviewer #2: Yes

2. Has the statistical analysis been performed appropriately and rigorously? 

Reviewer #1: Yes

Reviewer #2: Yes

3. Have the authors made all data underlying the findings in their manuscript fully available?

Reviewer #1: Yes

Reviewer #2: Yes

4. Is the manuscript presented in an intelligible fashion and written in standard English?

Reviewer #1: No

Reviewer #2: Yes

5. Review Comments to the Author

Reviewer #1: The paper is technically complex and hence challenging to resent.

I found the message was lost in the details.

It would be very helpful for the reader to have a "big picture" diagram as Figure 1.

Also, the author states that the magnification was 10x or 20X. It is more likely that a 10X or 20X objective was used and that is in combination with 10X binoculars so the magnification would be 100X and 200X. Please check this.

Reviewer #2: This study aimed to investigate the effect of TGF-β1 on intracellular signals associated with neurite outgrowth in the neuronal hybridoma cell line ND7/23. ND7/23 cell differentiated by addition of nerve growth factor. Cells extend neurites and express specific markers of sensory neurons. The authors used different methods to evaluate the impact of TGF-β1 on neurite outgrowth. TGF-β1 treatment inhibited neurite extension in differentiated ND7/23 cells and increased the phosphorylation of the collapsin response mediator protein-2 (CRMP2). Furthermore, both glycogen synthase kinase (GSK)-3β and Smad3 modulate phosphorylation of CRMP2 after treatment with TGF-β1.

The study provides evidence that TGF-β1 mediated inhibition of neurite outgrowth in sensory neurons appears to be regulated through a highly-conserved signaling pathway.

There are some minor points to be corrected:

1) Figure 1: Scale bar is missing in Figure 1A-C. Please insert one. Magnification indicators are not informative for the reader, please insert scale bars throughout the MS.

2) The authors write in the results’ section (Impact of TGF-β 1 on neurite outgrowth) “these cells were cultured with 50ng/ml rNGF for 1 day, the proportion of cells in Stage 1 decreased (two-tailed paired t-test for 0.5 versus 50 ng/ml rNGF conditions: t3=4.33, p=0.023), while the proportion of Stage 3 cells nearly doubled (two-tailed paired t-test: t3=3.58, p=0.037) – see pictures in Fig. 2E”. Figure 2E does not represent this statement. Please clarify.

3) Please indicate the significant difference in Figure 2D

4) What drug/treatment can block the effect of TGF-β1 in order to promote neurite outgrowth? Please discuss in detail. with specific details on molecular mechanisms.

6. PLOS authors have the option to publish the peer review history of their article (what does this mean?). If published, this will include your full peer review and any attached files.

Reviewer #1: No

Reviewer #2: No

---

## [Author Response · Author response to Decision Letter 0]

20 Oct 2020

Please see attached "response to reviews" word document. Here is what it contains:

We thank the two reviewers for their time and attention, as well as constructive comments on our manuscript. We have carefully addressed each of the comments in the revised documents. Our point-by-point responses are indicated in blue text below. Revisions made to the manuscript are also indicated in blue. We hope that you will now find our revised manuscript suitable for publication in PLOS One and look forward to your evaluation.

Reviewer #1: The paper is technically complex and hence challenging to read. I found the message was lost in the details. It would be very helpful for the reader to have a "big picture" diagram as Figure 1.

Response: we have added a new Figure1 that schematically summarizes the corneal wounding context in which we are working, and the specific signaling gaps we are attempting to fill. We have also modified the text of the introduction to more clearly motivate the present experiments, ending with the 2 specific questions we are asking about the role of GSK-3β and Smad signaling in mediating the anti-neuritogenic effects of TGF-β1 in sensory peripheral neurons such as those innervating the cornea.

Also, the author states that the magnification was 10x or 20X. It is more likely that a 10X or 20X objective was used and that is in combination with 10X binoculars so the magnification would be 100X and 200X. Please check this.

Response: since Reviewer 2 also brought up a related point, we now address both comments by providing scale bars on all photomicrographs presented in the Figures to this manuscript.

Reviewer #2: There are some minor points to be corrected:

1) Figure 1: Scale bar is missing in Figure 1A-C. Please insert one. Magnification indicators are not informative for the reader, please insert scale bars throughout the MS.

Response: as noted above in response to Reviewer 1’s second comment, we now provide scale bars on all photomicrographs presented in the Figures to this manuscript.

2) The authors write in the results’ section (Impact of TGF-β 1 on neurite outgrowth) “these cells were cultured with 50ng/ml rNGF for 1 day, the proportion of cells in Stage 1 decreased (two-tailed paired t-test for 0.5 versus 50 ng/ml rNGF conditions: t3=4.33, p=0.023), while the proportion of Stage 3 cells nearly doubled (two-tailed paired t-test: t3=3.58, p=0.037) – see pictures in Fig. 2E”. Figure 2E does not represent this statement. Please clarify.

Response: it is difficult, given the neuronal density in these cultures to capture pictures of sufficient resolution to show morphology, while also capturing a sufficient number of cells to illustrate the quantitative data in A-D, which was based on hundreds of neurons. We removed the statement “– see pictures in Fig. 2E” from the text, but also added labels to panel E of the Figure to indicate what stages the cells pictured were in. The only picture in which Stage 3 cells are visible is the one of a culture treated with rNGF. The proportion of Stage1 and 2 cells captured photographically is greater in cultures treated with Sema3A or TGF-beta. This is reflective of our quantitative data.

3) Please indicate the significant difference in Figure 2D

Response: done

4) What drug/treatment can block the effect of TGF-β1 in order to promote neurite outgrowth? Please discuss in detail. with specific details on molecular mechanisms.

Response: we have added a relatively speculative paragraph to the end of the Discussion, answering this question.

---

## [Editor Report · Decision Letter 1]

3 Nov 2020

How scars shape the neural landscape: key molecular mediators of TGF-β1’s anti-neuritogenic effects

PONE-D-20-16873R1

Dear Dr. Huxlin,

We’re pleased to inform you that your manuscript has been judged scientifically suitable for publication and will be formally accepted for publication once it meets all outstanding technical requirements.

Kind regards,

Antal Nógrádi, M.D., Ph.D., D.Sc.

Academic Editor

PLOS ONE
---

## [Editor Report · Acceptance letter]

11 Nov 2020

PONE-D-20-16873R1 

How scars shape the neural landscape: key molecular mediators of TGF-b1’s anti-neuritogenic effects 

Dear Dr. Huxlin:

I'm pleased to inform you that your manuscript has been deemed suitable for publication in PLOS ONE. Congratulations! Your manuscript is now with our production department. 

Kind regards, 

on behalf of

Prof. Antal Nógrádi 

Academic Editor

PLOS ONE